# Vitamin C Attenuates Oxidative Stress and Behavioral Abnormalities Triggered by Fipronil and Pyriproxyfen Insecticide Chronic Exposure on Zebrafish Juvenile

**DOI:** 10.3390/antiox9100944

**Published:** 2020-10-01

**Authors:** Madalina Andreea Robea, Roxana Jijie, Mircea Nicoara, Gabriel Plavan, Alin Stelian Ciobica, Carmen Solcan, Gilbert Audira, Chung-Der Hsiao, Stefan-Adrian Strungaru

**Affiliations:** 1Department of Biology, Faculty of Biology, Alexandru Ioan Cuza University of Iasi, Bd. Carol I, 20A, 700505 Iasi, Romania; madalina.robea11@gmail.com (M.A.R.); mirmag@uaic.ro (M.N.); gabriel.plavan@uaic.ro (G.P.); 2Marine Biological Station “Prof. dr. I. Borcea”, “Alexandru Ioan Cuza” University of Iasi, Nicolae Titulescu Street, No. 163, 907018 Agigea, Constanta, Romania; roxanajijie@yahoo.com; 3Department of Molecular Biology, Histology and Embryology, Faculty of Veterinary Medicine, University of Agricultural Science and Veterinary Medicine Ion Ionescu de la Brad, 8, Mihail Sadoveanu Alley, 700489 Iasi, Romania; carmensolcan@yahoo.com; 4Department of Chemistry, Chung Yuan Christian University, Chung-Li 320314, Taiwan; gilbertaudira@yahoo.com; 5Department of Bioscience Technology, Chung Yuan Christian University, Chung-Li 320314, Taiwan; 6Institute for Interdisciplinary Research, Science Research Department, Alexandru Ioan Cuza University of Iasi, Lascar Catargi Str. 54, 700107 Iasi, Romania; stefan.strungaru@uaic.ro

**Keywords:** neurotoxicity, vitamin C, insecticides, protective role, social behavior, autism spectrum disorder

## Abstract

Chronic exposure to synthetic insecticides in the early life of a child can lead to a series of disorders. Several causes as parental age, maternal smoking, birth complications, and exposure to toxins such as insecticides on childhood can lead to Autism spectrum disorder (ASD) occurrence. The aim of this study was to evaluate the potential protective role of vitamin C (Vit. C) from children’s supplements after 14 days chronic exposure to insecticide mixture fipronil (Fip) + pyriproxyfen (Pyr) on juvenile zebrafish for swimming performances, social behavior and oxidative stress associated with ASD model. Juvenile (14–17 mm) wild-type AB zebrafish (*Danio rerio*) (45 days) were exposed to relevant concentrations: vit. C (25 µg L^−1^), Fip (600 µg L^−1^/1.372 μM) + Pyr (600 µg L^−1^/1.89 μM), and [Fip (600 µg L^−1^/1.372 μM) + Pyr (600 µg L^−1^ /1.89 μM)] + vit. C (25 µg L^−1^). Our results showed that insecticides can disturb the social behavior of zebrafish during 14 days of the administration, decreased the swimming performances, and elevated the oxidative stress biomarkers of SOD (superoxide dismutase), GPx (glutathione peroxidase), and MDA (malondialdehyde). The vitamin C supplement significantly attenuated the neurotoxicity of insecticide mixture and oxidative stress. This study provides possible in vivo evidence to show that vitamin C supplements could attenuate oxidative stress and brain damage of fipronil and pyriproxyfen insecticide chronic exposure on zebrafish juvenile.

## 1. Introduction

Synthetic insecticides are closely related to human life since they have many applications in agriculture by protecting crop populations against insect species that carry diseases. Thus, children have contact with them before birth via contaminated food and water consumed by pregnant women and after birth by contacting with the environment directly. Unfortunately, chronic exposure to synthetic insecticides in the early life of a child can lead to a series of disorders. For example, prior studies found that several issues, such as parental age, maternal smoking, birth complications, and exposure to toxins, including insecticides, on childhood can lead to Autism spectrum disorder (ASD) occurrence [1,2]. In addition, there are a lot of risk factors that have specific actions depending on the period of exposure (prenatal, natal, and postnatal), duration, quantity, and the mode of intake of these insecticides [3].

Fipronil (Fip) is an insecticide used in veterinary products to kill fleas, ticks, and mites [4]. Most often, this product is formulated by mixing two or more chemicals. For instance, this insecticide is commonly combined with pyriproxyfen (Pyr), an insect growth regulator. Fip acts in the central nervous system (CNS) by blocking GABA (chemical messenger that is widely distributed in the brain) receptors which results in excess neuronal stimulation [5,6,7]. Considering the selectivity of this insecticide on perturbing GABA metabolism, it is intriguing to study the potency of this insecticide as an inductor for ASD in a zebrafish animal model. To the best of our knowledge, until now, there are no studies of Fip in modulating ASD symptoms in an animal model. However, the effects of Fip on wildlife organisms had been recorded in several prior studies [8]. Notochord degeneration, locomotor deficits, anxiety, changes in gene transcription, developmental abnormalities, and body mass reduction were several effects caused by Fip observed in zebrafish, fathead minnow, birds, and rodents [9,10,11,12,13,14,15,16].

Pyriproxyfen (Pyr) belongs to the class of juvenile hormone mimics, which targets the disruption of the endocrine system of insect eggs and larvae [17]. Even though it is used particularly for mosquitoes, its efficacy can also be seen against other species of insects and crustaceans [18]. In contrast to Fip, ecotoxicological data for Pyr effects on fish is still very limited. Several studies demonstrated that Pyr may lead to developmental deformities, changes in heart size, DNA damage, high levels of ROS (reactive oxygen species), enzyme inhibition, erratic swimming, and lethargy in *Xiphophorus maculatus*, *Danio rerio*, and *Haplosternum littorale* [17,19,20,21].

Autism spectrum disorder (ASD) is a series of neurobehavioral conditions which especially affect the communication and social domain of humans [22]. The incidence and prevalence of ASD have increased in recent years. In United States, the prevalence of ASD was estimated to be one in 54 children in 2016, and globally one from 588 children [23,24]. Its multifactorial character makes it harder to identify and diagnose properly in advance [25]. Many studies have been published in the past years seeking new therapies for preventing or ameliorating ASD-like behavior [26,27,28]. Besides drug treatments, some therapies including music therapy, acupuncture, massage, exercises, or animal-assistance led to significant improvements in the patients [27]. However, sometimes, these aforementioned methods are not sufficient for ASD conditions. Therefore, a combined intervention between drugs and complementary treatments would be more effective. Currently, ASD is defined on the basis of core deficits in social interaction and communication and repetitive and stereotyped symptoms [22,29]. At the moment, ASD does not have a treatment for its core; thus, each symptom presented in ASD people is treated individually [27,30,31].

Defined as an imbalance between antioxidants and pro-oxidants, oxidative stress (OS) has been often linked with ASD pathogenesis [32,33,34]. Reactive oxygen species (ROS) are generated through several pathways and a high level of ROS causes cell damage [35,36]. However, some defensive enzymes such as superoxide dismutase (SOD), glutathione peroxidase (GPx), and catalase (CAT) can counteract ROS production [36]. In addition, besides these enzymes, exogenous antioxidant supplementation has been associated with improvements in ASD symptoms [5,33,34,35].

Vitamin C (Vit. C), which is probably one of the most known vitamins, is implicated in many biochemical processes in the human body. Its activity in enzyme oxidation, immune function, and oxidative stress is often reported in the literature [37]. Even though it is a water-soluble vitamin, it helps vitamin E, a fat-soluble vitamin, through the regeneration of the active form, α-tocopherol [38], [39]. Vit. C is also required in the production of collagen, melanine, and carnitine [37,39]. Unlike most animals, humans lost the capacity to synthesize it endogenously [40]. Traditionally, vit. C deficiency has been called scurvy [39]. Scurvy manifestations include irritability, vascular fragility as a consequence of low collagen levels, dermatologic symptoms as gingival swelling, loss of the teeth, ecchymosis, infections, and extreme pain [39,41]. Interestingly, scurvy was also found to be linked with ASD in numerous case reports [42,43,44,45,46]. Parents frequently reported food selectivity in ASD children [47,48,49]. This highly selective eating behavior observed in ASD people raises awareness among clinicians due to the nutritional inadequacies which appear. Despite the fact that the human body is unable to produce vit. C, many experts believe that certain recommended supplementations can be effective. However, although it can be useful as an antioxidant agent in neutralizing free radicals and oxidants, due to the lack of studies, the potential of this vitamin in ASD is not fully elucidated [38,50].

*Danio rerio* (Hamilton, 1822), commonly known as zebrafish, are widely used as vertebrate models for prediction of pollutants (neuro)toxicity in aquatic systems due to their complex behavioral responses, short breeding cycle, transparent embryos, and high similarity with human genome [51,52]. Taken together, the aim of this study was to evaluate the possible protective role of vit. C after chronic exposure of insecticide mixture (fipronil and pyriproxyfen) on juvenile zebrafish swimming performances, social behavior, and oxidative stress associated with ASD model.

## 2. Materials and Methods

### 2.1. Ethical Note

The animals were strictly maintained and treated according to EU Commission Recommendation (2007) on guidelines for the accommodation and care of animals used for experimental and other scientific purposes, Directive 2010/63/EU of the European Parliament, and the Council of 22 September 2010 on the protection of animals used for scientific purposes. This experiment was approved by the Ethical Commission from the Faculty of Veterinary Medicine, University of Agricultural Sciences and Veterinary Medicine Iasi, with registration number 748/04.07.2019.

### 2.2. Chemicals

Superoxide Dismutase Determination Kit (SOD, 19160-1KT-F), Lipid Peroxidation (MDA) Assay Kit (MAK085-1KT), Glutathione Peroxidase Cellular Activity Assay Kit (GPx, CGP1-1KT), and Total Protein Kit, Micro Lowry, Peterson’s Modification (TP0300-1KT) were purchased from Merck, Germany. Physiological saline (0.90% NaCl) was purchased from Hemofarm (Romania). Ascorbic acid (Vit. C) was purchased from a local pharmacy in order to do a study in more related situations and conditions. The Vit. C was in a liquid form, soluble in water, and designated for children as a supplement. The composition of this supplement was acid ascorbic, glycerin, propylene glycol, and orange flavor. The supplement was diluted the desired concentration was obtained. For fipronil and pyriproxyfen (1:1), we used a common insecticide that is soluble in water and contains these two compounds with a quality certificate and it was purchased from a local store. Every 1.5 mL of this insecticide had the following composition: 67.5 mg of fipronil, 67.5 mg of pyriproxyfen, 0.3 mg of butylated hydroxyanisole, 0.15 mg of butylhydroxytoluene, and 60 mg of benzyl alcohol.

### 2.3. Animal Housing

A total of 60 juvenile (14–17 mm) wild-type AB (genetic lines) zebrafish (*Danio rerio*) (45 days) were used and housed in 90 L aquarium filled with dechlorinated water for 4 days accommodation. A standard laboratory condition with 26 ± 2 °C of temperature, and a pH of 7.5 with a 14:10 light:dark circuit (turned on at 9 a.m. and off at 7 p.m.) was maintained during the housing [53]. The juvenile zebrafish were fed two times per day with Tetramin Tropical Flakes (an average mean of 6.8 µg vit. C per fish per meal).

### 2.4. Experimental Design and Chronic Exposure

Before the experimental study was started, each group was evaluated in the swimming performance and social tests. These initial tests were conducted to demonstrate the normal behavior of each group before the treatment since previous studies shown that zebrafish had a high behavioral variability between each specimen which resulted in significant differences between the untreated groups [54,55,56]. This test was done after the accommodation period from 10 a.m. to 8 p.m. and repeated for 4 days. A number of 15 specimens were separated in four 10 L experimental aquariums with dechlorinated tap water. The grouping was as follows: Group 1 (the control group), Group 2 (25 µg L^−1^ vit. C), Group 3 (600 µg L^−1^ Fip + 600 µg L^−1^ Pyr), and Group 4 ([600 µg L^−1^ Fip + 600 µg L^−1^ Pyr] + 25 µg L^−1^ vit. C). Every day, the water from aquariums was replaced and the concentrations of the tested insecticides was maintained. Regarding the vit. C exposure, the fish were removed from the tanks and exposed to vitamin before being returned to the tank. The vit. C was administrated daily in the morning, in single used 150 mL PE (polyethylene) cups, for 30 min exposure in a solution (vit. C and dechlorinated tap water). This was done according to other studies that investigated the vit. C effects on zebrafish [57]. The exposure experiment was conducted for 14 days. The entire experimental design is presented in Figure 1 and was conducted in triplicates.

### 2.5. Behavioral Assessment

The social interaction test was performed daily to measure the changes produced by insecticide exposure and the possible positive effect of vit. C on behavioral endpoints. The test was performed in a multipurpose T maze (10 × 50 × 50 cm/height × length × width), filled with water to a height of 5 cm. The animals were allowed to acclimate to the experimental area for a 30 s period and then their behaviors were recorded by using a professional camera located above the maze over a period of 4 min. In order to quantify fish social preference, the experimental chamber was divided into 3 zones: left arm with the social stimuli formed from 4 group colleagues that were systematically replaced with those that were tested, central area, which considered as the start point, and right arm, where was empty and representing the antisocial area. The amount of time spent by the experimental fish on each zone was measured using EthoVision XT 11.5 software that was previously calibrated for the social test (NOLDUS, Holland). Afterwards, the same T maze was modified to measure the daily swimming performance of test fish where the subject had no stimulus. The length of this test was 4 min for the subjects to explore the entire maze. EthoVision XT 11.5 that was previously calibrated for this test and used to measure the performance variables like total distance swam (cm), swim velocity (cm s^−1^), and anxiety level (based on the subject movements/freezing time). Data were presented as average ± SD.

### 2.6. Oxidative Stress Evaluation

After the chronic exposure, each fish (15 specimens per experimental group) was well homogenized in 10 volumes of ice-cold saline (0.90% NaCl) and centrifuged at 5500 rpm for 10 min at 4 °C according to the previously reported protocol [58,59]. The supernatant of each sample was transferred to a clean PE tube and was divided into aliquots for total protein analysis, SOD, GPx, and MDA measurement. The SOD, GPx, and MDA levels, as well as the protein concentrations of tissue suspensions, were determined using the assay kits from Merck. Each parameter analyzed was measured following Merck protocols. A spectrophotometer (Specord 210 Plus producer Analytik Jena, Germany) was used to determine total protein, SOD, GPx, and MDA. Data were expressed as average ± SD.

### 2.7. Statistical Analysis

Firstly, we analyzed the normality and distribution of the data with the Shapiro-Wilk test. Later, the results obtained after this test proved that all of the data sets of the experimental groups were normally distributed. Afterwards, the one-way ANOVA followed by the Tukey HSD (honestly significant difference) test were performed to demonstrate the significant different variance of the investigated behavioral variables from the initial condition of the subjects (initial behavior) to the end of the treatment in the case of each group [52,55]. In the case of oxidative stress, the variance was calculated with multiple comparisons between control and treatment groups. The statistical analyses were conducted with OriginPro v.9.3 (2016) software that was designed and created by OriginLab Corporation, Northampton, MA, USA.

## 3. Results

### 3.1. The Effects of Vitamin C in Insecticide Mixture on Swimming Performances

Zebrafish juvenile treated with 25 µg L^−1^ vit. C showed an increase in locomotor activity parameters (Figure 2). This was evaluated by measuring specific parameters like total distance and swimming velocity. Generally, the time spent as moving/freezing represents the anxiety level [55,56]. From the results, the control group did not show significant changes for the total distance traveled between pretreatment days and exposure period (691.05 ± 64.2 cm vs. 719.66 ± 84.9 cm, *p* > 0.05 ANOVA). However, a significant difference was observed for the total distance swam (*p* < 0.05 ANOVA) in the group with a vitamin supplement. This group exhibited the highest value of total distance traveled in day 11 (1193.43 ± 107.2 cm, *p* < 0.001 Tukey) compared to pretreatment days (758.06 ± 65.58 cm), followed by day 5 (1126.14 ± 66.3 cm) and day 8 (1109.51 ± 93.9 cm). On the contrary, the insecticide mixture group presented a significant descending trend in distance swam (*p* < 0.05 ANOVA) between pretreatment days and chronic exposure (674.8 ± 46.1 vs. 530.2 ± 57.63 cm). The lowest values were recorded in day 13: 499.3 ± 62.01 cm (*p* < 0.01 Tukey) and day 14: 481.8 ± 68.7 cm (*p* < 0.01 Tukey). Meanwhile, simultaneous chronic exposure to insecticide mixture and vit. C decreased the distance swam in the first several days but it was not significant for the entire experiment (pretreatment days: 769.9 ± 53.9 cm vs. chronic exposure: 707.4 ± 79.1 cm, *p* > 0.05 ANOVA) (Figure 2A).

Next, the swim velocity parameter recorded similar modifications to the total distance swim parameter. Here, the control group had a velocity between 2.90 ± 0.36 cm s^−1^ – 3.25 ± 0.35 cm s^−1^ without significant changes during the experiment (*p* > 0.05 ANOVA). Furthermore, velocity parameter showed significant changes for vitamin group in day 6 (4.40 ± 0.34 cm s^−1^, *p* < 0.05 Tukey), day 7 (4.41 ± 0.35 cm s^−1^, *p* < 0.05 Tukey), day 8 (4.48 ± 0.41 cm s^−1^, *p* < 0.05 Tukey), day 11 (4.66 ± 0.50 cm s^−1^, *p* < 0.05 Tukey), and day 14 (4.32 ± 0.42 cm s^−1^, *p* < 0.05 Tukey). On another hand, the group exposed to insecticide mixture had a strong shock after the intake of 600 µg L^−1^ Fip + 600 µg L^−1^ Pyr in the first day of the chronic exposure (1.60 ± 0.29 cm s^−1^, *p* < 0.01 Tukey) compared to pretreatment days (3.08 ± 0.29 cm s^−1^) (Figure 2B). Meanwhile, the group was exposed to insecticide mixture and vit. C had a similar trend to control group, without recording important changes between pretreatment and chronic exposure (pretreatment: 3.21 ± 0.22 cm s^−1^ vs. 2.80 ± 0. 50 cm s^−1^, *p* > 0.05 ANOVA).

The behavioral trend for anxiety level was related to the swimming velocity and total distance traveled. The results showed no significant modifications observed in the control group (pretreatment: 184.3 ± 6.28 s vs. exposure period: 181. 1 ± 15.8 s, *p* > 0.05 ANOVA). Furthermore, the group that was exposed to 25 µg L^−1^ vit. C had the highest values for the moving time spent compared to pretreatment days (185.3 ± 8.46 s) in day 5 (216. 2 ± 8.63 s, *p* = 0.01), day 6 (218.4 ± 8.42 s, *p* < 0.05 Tukey), day 10 (215.4 ± 6.87 s, *p* < 0.05 Tukey), day 12 (219.7 ± 8.97 s, *p* < 0.05 Tukey), and day 14 (220.9 ± 6.32 s, *p* < 0.05 Tukey). Meanwhile, the insecticide mixture had the highest impact on the first day of administration (pretreatment: 190.7 ± 10.5 s vs. day 1: 95.3 ± 18.1 s, *p* < 0.01 Tukey). After day 1, the group started to move more by recording some high peaks in day 9 (162.5 ± 16.1 s), day 10 (161.2 ± 12.2 s), day 11 (161.2 ± 16.7 s), and day 12 (169.7 ± 16.5 s). In the last 2 days of the experiment (day 13: 134.7 ± 10.4 s, *p* < 0.01 Tukey and day 14: 148.7 ± 13.4 s, *p* < 0.05 Tukey), the group spent less time moving compared to pretreatment days (Figure 2C). However, treatment with vit. C influenced the moving time spent parameter of the group that received insecticide mixture. The results during the administration period (day 4: 187.3 ± 9.12 s, day 9: 191.3 ± 12.3 s, day 11: 186.4 ± 11.2 s, day 14: 186.1 ± 16.9 s) were closed to those from pretreatment (190.6 ± 6.74 s) (*p* > 0.05 Tukey).

Further, there were no significant changes in the case of the inactive status parameter for the control group (*p* > 0.05 ANOVA) while the administration of vit. C supplement led to a decrease in time spent in inactivity for group 2 (pretreatment days: 57.8 ± 5.78 s vs. exposure period: 36.4 ± 7.45 s). Significant results were observed in day 1 (32.3 ± 5.28 s, *p* < 0.01 Tukey), day 2 (35.4 ± 4.40 s, *p* < 0.01 Tukey), day 11 (15.4 ± 4.58 s, *p* < 0.01 Tukey), day 12 (19.3 ± 8.91 s, *p* < 0.01 Tukey), day 13 (34.9 ± 12.2 s, *p* < 0.01 Tukey), and day 14 (15.8 ± 5.04 s, *p* < 0.01 Tukey). Meanwhile, the lowest value of this parameter was recorded for the group exposed to the insecticide mixture (pretreatment 46.3 ± 7.42 s vs. chronic exposure 94.8 ± 12.7 s, *p* < 0.01 Tukey) which suggested an increase of the anxiety level caused by the neurotoxicity effects. By far, as can be seen in Figure 2D, the group exposed to the insecticide mixture and vitamin C supplement had a lower level of freezing time, which promotes the beneficial role of neurologic protection against the harmful effects.

### 3.2. Effect of Vitamin C on Social Behavior

Social interaction test was conducted to study the effect of insecticides and their relationship with vit. C in zebrafish social behavior (Figure 3). Our results showed that the control group spent most of the time in the left arm (pretreatment days: 124.7 ± 13.3 s vs. exposure period: 119.6 ± 15.5 s) compared to right arm (pretreatment days: 54.7 ± 13.9 s vs. exposure period: 51.8 ± 12.1 s) and center arm (pretreatment days: 51.4 ± 5.60 s vs. exposure period: 58.1 ± 7.93 s) (Figure 3A). As we expected, zebrafish juveniles treated with vit. C responded well to the social stimuli from the left arm (pretreatment days: 132.2 ± 11.8 s vs. chronic exposure: 120.5 ± 15.3 s) and had the same pattern as the control group for the center arm (pretreatment days: 56.5 ± 8.05 s vs. chronic exposure: 57.6 ± 7.29 s) and the right arm (pretreatment days: 49.7 ± 6.23 s vs. exposure period: 45.4 ± 9.16 s) (Figure 3B). On the contrary, insecticides disturbed the social behavior of zebrafish during 14 days of administration. This phenomenon was supported by the significant decrease in time spent next to the group area (pretreatment days: 119.8 ± 12.5 s vs. chronic exposure: 53.6 ± 11.09 s, *p* < 0.001), indicating the zebrafish preference to the right arm (pretreatment days: 63.8 ± 12.7 s vs. chronic exposure: 98.3 ± 15.1 s, *p* < 0.001) (Figure 3C). Interestingly, when vit. C and insecticides mixture were given simultaneously, zebrafish presented a descending trend regarding the time spent in the group zone (pretreatment days: 145.8 ± 11.5 s vs. chronic exposure: 105.4 ± 17.7 s, *p* = 0.001) and an increase in time for right arm (pretreatment days: 69.8 ± 9.69 s vs. chronic exposure: 82.5 ± 12.1 s, *p* < 0.001). However, in some time of exposure, a higher time spent in the left arm was still observed in this group, which was not displayed by the insecticides group (Figure 3D). Taken together, the results demonstrated that at the end of the experiment, vit. C supplement could be implicated in reducing the neurotoxicity of the insecticides in terms of zebrafish social behavior response, which might indicate the protective function of this vitamin against brain damage.

### 3.3. Oxidative Stress

In the present study, we investigated the superoxide dismutase (SOD), glutathione peroxidase (GPx), and lipid peroxidation (MDA) activities in order to assess the oxidative stress in the zebrafish juveniles, as the result of the chronic exposure of insecticide mixture and vit. C supplements to reduce the harmful effects of the insecticides. SOD and GPx are considered to be the first line of defense against oxidative stress since SOD helps in conversion of O_2_^−^ to O_2_ and H_2_O_2_ while GPx can effectively catalyze the reduction of H_2_O_2_ to water and a variety of organic peroxides to alcohols [52]. As can be seen in Figure 4, all three biochemical indicators of the oxidative stress resulted in a significant variance (*p* < 0.05 ANOVA). Furthermore, the administration of vit. C supplement on zebrafish juveniles had significant antioxidants effects on SOD activity. The group that received vit. C had a significantly lower (*p* < 0.05 Tukey) concentration of SOD (2.04 ± 0.18 units of SOD (USOD)/mg protein) than the control group (2.536 ± 0.172 USOD/mg protein) at the end of the experiment, demonstrating the antioxidant role in the body. Meanwhile, the group exposed to the insecticide mixture (Fip + Pyr) had a significantly high (*p* < 0.001 Tukey) values of SOD (4.82 ± 0.35 USOD/mg protein) compared to the control group. As we expected, this level of SOD was significantly higher than the group treated with the only vit. C and insecticide mixture + vit. C. Taken together, this result proved that in the case of SOD activity, vit. C supplement significantly reduced the toxicity level of insecticide, even though it was not enough to reach the same level as the untreated group. In addition, a similar phenomenon was also observed in the case of GPx and MDA. In GPx measurement result, vit. C supplement significantly decreased the oxidative stress produced by the insecticide mixture (Fip + Pyr: 0.18 ± 0.03 UGPx/mg protein vs Vit. C + [Fip + Pyr]: 0.09 ± 0.023 UGPx/mg protein) even though it was still slightly higher than the control group (0.032 ± 0.006 UGPx/mg protein). Furthermore, similar results were observed in their MDA activity level where the oxidative stress had the highest value in the insecticide mixture group and it was significantly decreased with the action of vit. C.

## 4. Discussion

Insecticides are world widely used and, most of them, are acting on the CNS of insects [4]. One of the common insecticides, Fipronil, is known for its action on GABA-mediated chloride channels by inhibiting the binding of GABA to its receptor [4,60]. Toxicological findings suggest that Fip can induce developmental deformities, cognitive impairments, and behavioral changes [7]. In zebrafish, 0.4 to 0.33 mg L^−1^ Fip can lead to notochord degeneration, locomotor defects, spine bending, anxiety, and swimming alterations [11,15,61]. Furthermore, when mixed with another insecticide (200 µg L^−1^ Fip and 50 mg L^−1^ buprofezin), its behavioral effects in common carp (*Cyprinus carpio*) are represented by erratic swimming, rapid opercular beats, loss of equilibrium, and air gulping after 96 h intake [62]. What needs to be taken into consideration is the fact that chemicals may have synergistic or antagonistic effects when they are presented in a combination. Based on this fact, analyzing the toxicant-related changes in behavior must be done. In addition, as an indicator of chemicals toxicity in fish that occurs earlier than death, swimming behavior is often evaluated [63].

In our study, we used an insecticide mixture composed of Fip and Pyr as this solution is commonly found in veterinary products. Decreases in distance swam, average velocity, and time spent moving were observed in juvenile zebrafish after 14 days of exposure to Fip and Pyr. In line with these results, similar results were noticed at shrimp after 29 days at 0.1 and 1.0 µg L^−1^ Fip. The effects of this pesticide were including circle and involuntary movements, inactivity, and death of the shrimps [64]. Moreover, short-term exposure (24 h) to concentrations higher than 142 µg L^−1^ Fip significantly impaired the swimming performance of fathead minnow (*Pimephales promelas*) [12]. In addition, another prior study also found that Pyr caused abnormal swimming behavior, erratic swimming, loss of equilibrium, and lethargy in southern platyfish (*Xiphophorus maculatus*) incubated with 10 and 20 µg L^−1^ of Pyr for 24 and 72 h [19]. However, no significant effects on distance swam, velocity, and anxiety-like behavior were seen in zebrafish after administration of 0.125, 0.675, and 1.75 mg L^−1^ Pyr. Although, disruptions of inhibitory avoidance memory were present at each concentration [21]. A single administration of Pyr at 10 and 100 µg L^−1^ during 21 days caused endocrine and reproductive alterations in zebrafish. 

An altered expression of hypothalamus-pituitary-gonadal axis genes was also recorded for both sexes [65]. Furthermore, embryological exposure of zebrafish to Pyr ended with a high mortality rate and craniofacial deformities at concentrations higher than 5.2 µM [66]. A recent study indicated that toxicity impact induced by Pyr might be associated with reduced levels of cortisol in a zebrafish experiment. Cortisol is known as a hormone that regulates stress response and its increase proves the presence of a stressor in the environment [67]. Adult zebrafish exposed to 0.125, 0.675, and 1.75 mg L^−1^ Pyr showed decreased cortisol concentrations after 96 h [11]. Pyr may have the capacity to modify the function of the endocrine system. Additionally, the activity of some structures from successive negative contrast (SNC) can be impaired as a result of Pyr influence in promoting important alterations of behavior and cognition [11,68]. Regarding a Fip effect on cortisol level, it was showed that administration of 0.0021 and 0.0042 mg L^−1^ of Fip for 96 h through water and food can increase cortisol level in Nile tilapia (*Oreochromis niloticus*) [69]. Based on these previous findings, we can conclude that Fip with Pyr may have a synergistic effect on affecting zebrafish swimming performance since in the current study; the combination of these insecticides caused more severe effects.

To the best of our knowledge, this is the first study that investigated the neuroprotective role of vit.C on zebrafish juveniles against the harmful effects of chronic exposure of fipronil and pyriproxyfen, two common insecticides that are used daily by the human population. In addition, to mimic a real-life scenario, we used vit. C from infant supplements in the current research. Known as an antioxidant, vit. C has been reported to be used as a potential treatment therapy in ASD individuals [27,28,70]. A double-blind, placebo-controlled study assessed the efficiency of vit. C in reducing autism severity in a group of 18 adolescents after giving a dose of 8 g/70 kg/day measured through the Ritvo-Freeman scale [71]. Similar to this study, another randomized, double-blind, placebo-controlled three-month study showed that supplementation with vit. C (600 mg kg^−1^) can reduce the ostium secundum (OS) in ASD individuals [72]. However, a recent randomized, controlled 12-month trial made on ASD people concluded that administration of 500 mg kg^−1^ vit. C cannot improve ASD symptoms [73]. Unfortunately, the lack of sustained results for vit. C supplementation in ASD is significant and causes for the rise of ASD throughout the population remain unknown. The possible contributing factor to an ASD incidence is the introduction of chemicals with certain actions into the environment [64]. These substances may influence individual health and sometimes those effects are irreversible. Up to now, studies in vit. C role in reducing the toxicity level of pesticides on fish is not common and even though we identified a few of them, no one investigated the fipronil and pyriproxyfen neurotoxicity. For example, Ghazanfar et al. exposed adults of carp fish (*Cyprinus carpio*) to an insecticide (from Pakistan market) mixture of fipronil (200μg L^−1^) and buprofezin (50 mgL^−1^) for three weeks [62]. In their study, it was investigated the effects of low (25 mgL^−1^) and high doses (50 mg L^−1^) of pure solid vit. C that was dissolved in the medium. Later, the authors measured the SOD activity in the liver, kidney, brain, and gills of the fish. In contrast with the current results, in all of the organs, the highest values (unit/mg protein) of SOD were reported in the control group while the lowest value was displayed in the insecticide-treated group. Later, when vit. C was administrated; it was increased until it was in a higher level than the control group in some cases. In addition, based on other presented results, vit. C supplementation in water reduced the mortality, the genotoxicity, and biochemical damage resulted from fipronil and buprofezin exposure. Meanwhile, in the case of SOD and GPx, zebrafish were found to adopt a mechanism to compensate for the oxidative damage induced by deltamethrin, another insecticide that is also neurotoxic, [52,74] resulting in an increment of their values. This mechanism is active in the embryonic stage of development for zebrafish [17]. However, the administration of vit. C at the same time showed its capacity to neutralize ROS production through stimulation of SOD, GSH, and GST activity, which was in line with the current study [75].

The social behavior test that was conducted in this study has been applied in the research of ASD. In zebrafish, a survival rate of zebrafish in the environment depends on social bound in shoal formation. In a normal condition, it is attracted by social stimuli that come from individuals of the same group. The tendency of an animal to interact and to approach a conspecific is well conserved in zebrafish [76], thus, the zebrafish avoidance behavior from social stimuli becomes a suitable model to study ASD when a new treatment for human patients is developed [77]. To assess the social preference behavior of zebrafish juveniles, we used individuals from its group as a stimulus. Simultaneous administration of 600 µg L^−1^ Fip and 600 µg L^−1^ Pyr during 14 days conducted to a reduction in social interaction in juvenile zebrafish. Consequently, this disruption in the sociability of juveniles was improved by vit. C exposure. Furthermore, this insecticide mixture might induce alterations in various brain regions causing cell malfunction and death. As we know, Fip metabolites can block GABA gated chloride channels in insects. In bees, Fip played a role in motor activity reduction and perturbations in bee behaviors such as seizures, tremors, agitation, and paralysis [78]. However, the specific mechanism of its action in fish and rodents is not fully understood. In a rodent study, it was shown that concomitant administration of 10 mg kg^−1^ Fip and 1 mg kg^−1^ picrotoxin, a GABA antagonist, have a synergistic effect on rodents’ memory formation and consolidation after 15 days [79]. Furthermore, some prior studies revealed the importance of excitatory glutamate and inhibitory GABA relation in brain development and functioning. Its imbalance is often linked with neurodevelopmental disorders such as attention deficit, hyperactivity disorder, epilepsy, and ASD [80,81]. Synthetized from glutamate, the GABA neurotransmitter is implicated in action potentials decrease by regulating the activity of the neurons [82,83,84]. Neuronal migration, synaptogenesis, differentiation, and growth of neuronal cells are controlled by glutamate, the primary excitatory neurotransmitter of CNS [84]. There are multiple studies that report elevated glutamate and low GABA levels in ASD individuals [82,85,86,87]. Thus, in the present study, changes in social behavior caused by the insecticide mixture were similar to those observed in an ASD model. Finally, our study demonstrated the influence of vit. C in ameliorating behavior alterations and oxidative stress products that were triggered by the mixture of two commonly used insecticides (Figure 5). Although, we showed here significant results for vit. C supplementation, there are also several limitations of our study as: the lack of measurements regarding quantification of vit. C in water and fish tissues. With other words, these effects of insecticides are significant and sometimes they alter the body systems in the long term. These events are connected and if just one is disrupted, this cycle is broken, e.g., intake of chemicals. Clearly, the complexity of ASD and the multitude of events that occur must be further studied through experimental research. 

## 5. Conclusions

Fipronil and pyriproxyfen used in the studied concentrations had significant toxic effects on juvenile zebrafish brain and behavior. They triggered a high increase in the intensity of the antisocial behavior that is similar to a zebrafish ASD research model. Furthermore, oxidative stress was also affected by these insecticides. However, vit. C supplements significantly reduced the neurotoxicity of insecticide mixture and oxidative stress. In the case of SOD, the group was treated with vit. C had a lower level of oxidative stress compared with the control group suggesting the antioxidant efficiency in protecting the body. In this study, even though the toxicity of these insecticides was not completely neutralized by vit. C administration, it significantly helped the zebrafish juvenile detoxification mechanism. Thus, the present study highlighted a potential application of vit. C supplementation, which is generally used for the brain protection of infants and children, against a daily chronic exposure of insecticides which may result in irreversible damage. Nevertheless, the need for more studies regarding vit. C efficiency is necessary. In future studies, more efforts to better understand the cellular protection mechanisms and to investigate more concentrations are worth exploring.

## Figures and Tables

**Figure 1 antioxidants-09-00944-f001:**
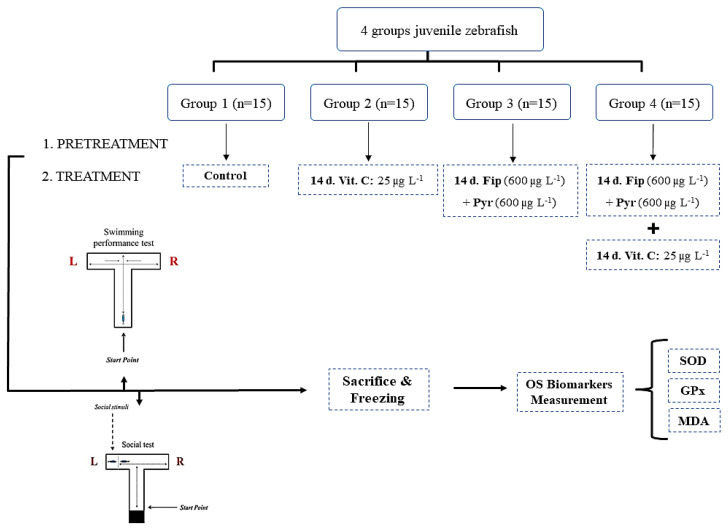
Schematic diagram of insecticide mixture and vitamin C administration. (d: days, Fip: fipronil, GPx: glutathione peroxidase, MDA: malondialdehyde, OS: oxidative stress, Pyr: pyriproxyfen, SOD: superoxide dismutase, Vit. C: vitamin C).

**Figure 2 antioxidants-09-00944-f002:**
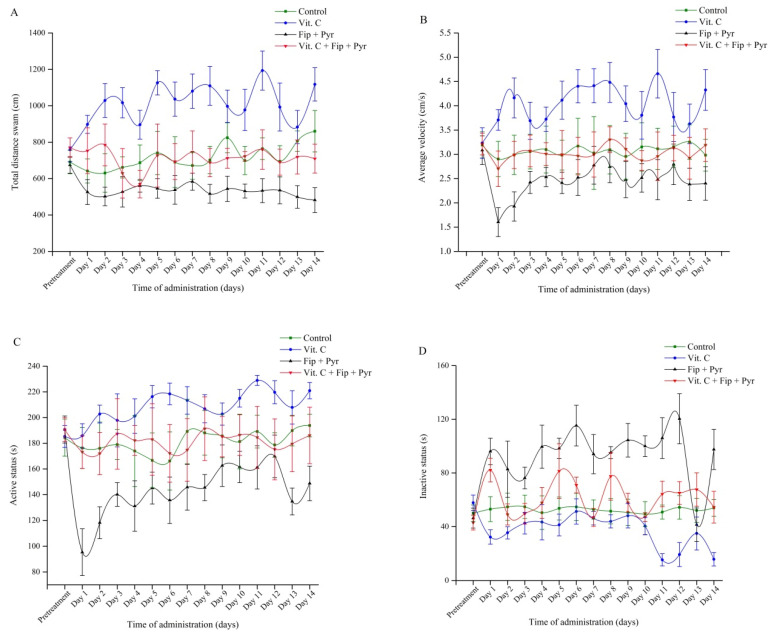
Zebrafish juvenile swimming performance test results during the experiment (*n* = 45 per group combined with triplicates experiments). (**A**) Total distance swam (cm). (**B**) Average velocity (cm/s). (**C**) Active status (s). (**D**) Inactive status (s). Green: control group, Blue: Vit. C group (14 d. 25 µg L^−1^), Black: insecticide mixture group (600 µg L^−1^ Fip + 600 µg L^−1^ Pyr), Red: insecticide mixture and vit. C group (14 d. [600 µg L^−1^ Fip + 600 µg L^−1^ Pyr] + 25 µg L^−1^ vit. C). The groups were compared to pretreatment days and the results were represented as average ± SD.

**Figure 3 antioxidants-09-00944-f003:**
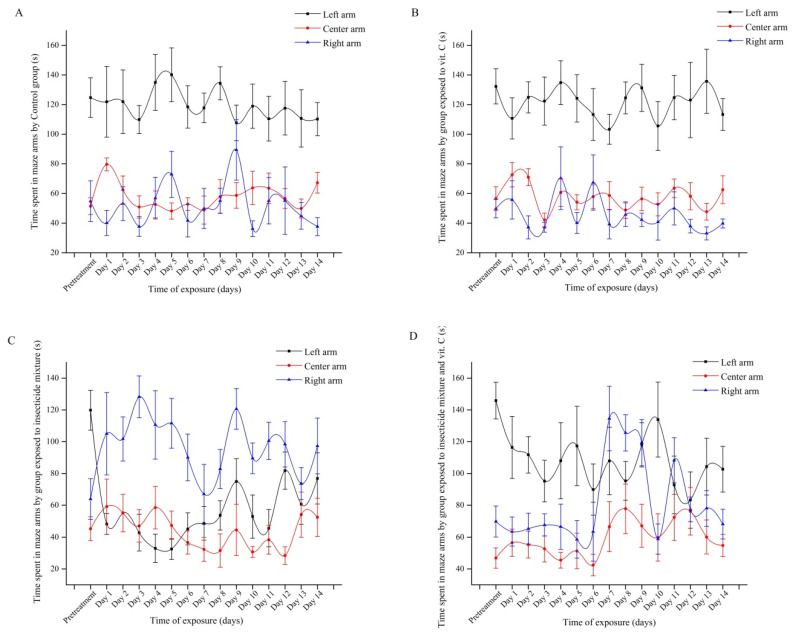
Time spent in maze arms during social interaction test by (**A**) control group and group exposed to (**B**) vit.C (14 d. 25 µg L^−1^), (**C**) insecticide mixture (600 µg L^−1^ Fip + 600 µg L^−1^ Pyr), and (**D**) insecticide mixture and vit.C (14 d. [600 µg L^−1^ Fip + 600 µg L^−1^ Pyr] + 25 µg L^−1^ vit. C) (*n* = 45 per group). Black: left arm, Red: center arm, and Blue: right arm. The groups were compared to pretreatment days and the results were represented as average ± SD.

**Figure 4 antioxidants-09-00944-f004:**
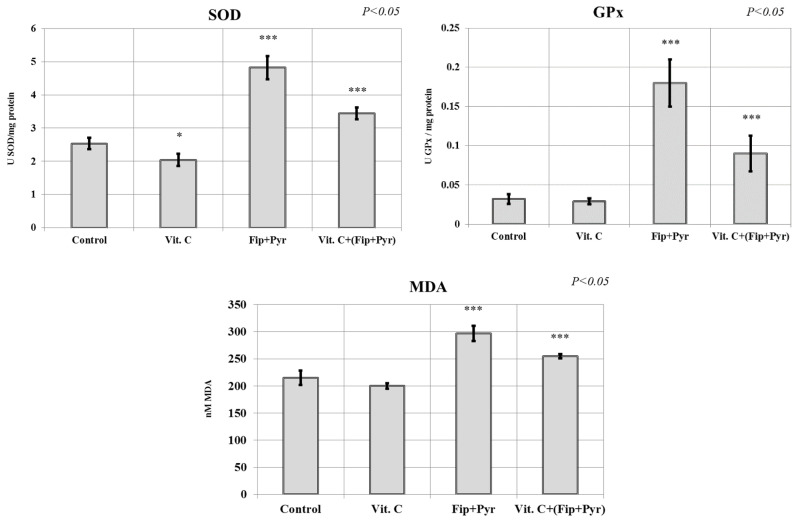
The activity of superoxide dismutase (SOD), glutathione peroxidase (GPx), and lipid peroxidation (MDA) levels in zebrafish chronically exposed to the chemicals. Data were expressed as average ± SD (*** *p* < 0.001 Tukey compared with control, * *p* < 0.05 Tukey compared with control, *p* < 0.05 ANOVA).

**Figure 5 antioxidants-09-00944-f005:**
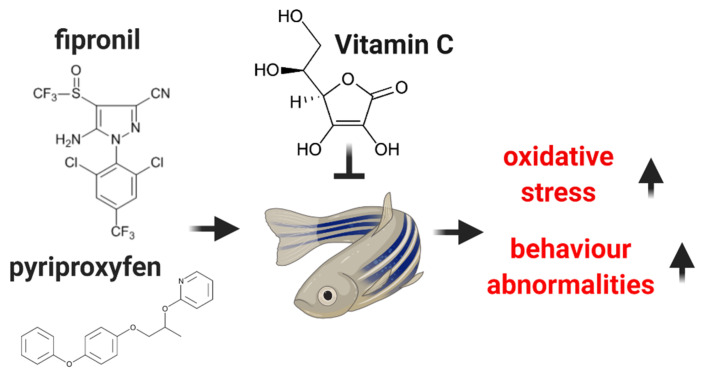
Schematic diagram of this experimental study. The juvenile zebrafish were exposed to a mixture of fipronil and pyriproxyfen. The insecticides triggered oxidative stress and behavioral abnormalities in zebrafish. However, these adverse effects were attenuated by vitamin C, which was showed by the reduction of oxidative stress marker expression in the brain and restoration of behavioral abnormalities in swimming performance and social interaction.

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
