# Peer review of "Vitamin C Attenuates Oxidative Stress and Behavioral Abnormalities Triggered by Fipronil and Pyriproxyfen Insecticide Chronic Exposure on Zebrafish Juvenile"

_antioxidants, 2020, doi:10.3390/antiox9100944_

Round 1
Reviewer 1 Report
The authors presented a revised version of the manuscript I have previously reviewed.
All my major queries have been fully addressed.
There are just some minor inconsistencies which can be even corrected during the Proofreading process:
- Line 38: in vivo must be written in Italics.
- Line 385: the semicolon must be removed.
By consequence, the manuscript can be accepted for publication in its current form.
Author Response
Reviewer 1:
- Line 38: in vivo must be written in Italics.
- Line 385: the semicolon must be removed.
We made the changes in the manuscript as the reviewer kindly suggested.
Reviewer 2 Report
Upon revision, the manuscript is improved. However, as described in the prior review of this manuscript, the authors have not demonstrated that vitamin C is being internalized by the exposed zebrafish and altering vitamin C status. Also, some of the details provided in the response to review must be included in the manuscript. More specifically:
- The introduction still does not address manuscripts vitamin C use in zebrafish models and/or effects against toxins.
- Line 142: add the vitamin C content of the diet here
- Line 153-154: Add more details about vitamin C exposure. The inference here is that fish were removed from the tank and exposed to ascorbic acid for 30 minutes before being returned to the tank. Change this to be clear.
- Along with above, please note if the contents of the cup (the remaining ascorbic acid) were also added back to the tank.
- Line 364: Do not refer to vitamin C as "C". Also, this is not relevant to a human condition since humans do not bathe in vitamin C solutions. Please remove or revise this sentence.
- Line 407: You present no evidence of vitamin C "intake". "Exposure" is probably the only word you could use here.
- Line 440: "Vit. C supplements" suggests dietary. There is no evidence that fish accumulated vitamin C through the digestive tract.
- In the discussion: The lack of measurement of vitamin C in the fish or the water needs to be presented as a limitation of the study. Even if this is to be added in future work, the interpretation of data presented is limited by a lack of vitamin C quantification.
- Vitamin C is unstable in neutral solutions and can spontaneously oxidize in fish water (resulting in significantly lower levels in a matter of minutes). The result could be the generation of superoxide and/or hydrogen peroxide. Please discuss this in light of your exposure regime. Is there any known research about the interaction with any of these redox compounds with the insecticides used?
Author Response
Reviewer 2:
- The introduction still does not address manuscripts vitamin C use in zebrafish models and/or effects against toxins.
We understand your opinion but we were focused in the current study to observe the capacity of vit. C use in combating reactiv oxygen species (ROS) after administrating an insecticide mixture to zebrafish. It is well-known that oxidative stress has a great impact on brain functioning: ”oxidative stress (OS) has been often linked with ASD pathogenesis [32–34].”. Due to the lack of therapies for ASD individuals, we started to investigate possible beneficial effects of vit. C in this domain by studying it through zebrafish, a popular organism model in research. We have designed a study by using insecticides to which humans are daily exposed and vit. C as an alternative solution against ROS. Beside this, we also quantified the social impairments, one of the ASD specific mark.
- Line 142: add the vitamin C content of the diet here.
As we mentioned in the last cover letter the content of vit C. was an average mean of 6.8 µg vit. C per fish per meal. We also added in the manuscript as the reviewer indicated here ” The juvenile zebrafish were fed two times per day with Tetramin Tropical Flakes (an average mean of 6.8 µg vit. C per fish per meal).”
- Line 153-154: Add more details about vitamin C exposure. The inference here is that fish were removed from the tank and exposed to ascorbic acid for 30 minutes before being returned to the tank. Change this to be clear.
As the reviewer suggested we added more information about vitamin C administration in the manuscript as: ”Regarding the vit. C exposure, the fish were removed from the tanks and exposed to vitamin before being returned to the tank. The vit. C was administrated daily in the morning, in single used 150 ml PE cups, for 30 minutes exposure in a solution (vit. C and dechlorinated tap water).”.
- Along with above, please note if the contents of the cup (the remaining ascorbic acid) were also added back to the tank.
The contents of the cup were not added back to the tank because we transferred the animals with fishing nets from cups to the specific tanks to avoid vitamin surplus.
- Line 364: Do not refer to vitamin C as "C". Also, this is not relevant to a human condition since humans do not bathe in vitamin C solutions. Please remove or revise this sentence.
We replaced ”C” with ”vit. C” and removed some parts of the sentence as the reviewer indicated here and the final form was ” In addition, to mimic a real-life scenario we used in the current research vit. C from infant supplements.”.
- Line 407: You present no evidence of vitamin C "intake". "Exposure" is probably the only word you could use here.
We replaced the word as the reviewer indicated: ” Consequently, this disruption in the sociability of juveniles was improved by vit. C intake” with ”Consequently, this disruption in the sociability of juveniles was improved by vit. C exposure.”
- Line 440: "Vit. C supplements" suggests dietary. There is no evidence that fish accumulated vitamin C through the digestive tract.
In our study, we did not quantified vitamin C accumulation, this being one of our limitations regarding this study. Due to this, we suggested possible beneficial effects of vitamin C administration as a result of vitamin exposure, mainly based on the oxidative stress parameters and then behavioural abnormalities: ”that in the case of SOD activity, vit. C supplement significantly reduced the toxicity level of insecticide, even though it was not enough to reach the same level as the untreated group. In addition, a similar phenomenon was also observed in the case of GPx and MDA. In GPx measurement result, vit. C supplement significantly decreased the oxidative stress produced by the insecticide mixture (Fip+Pyr: 0.18±0.03 UGPx/mg protein vs Vit. C+[Fip+Pyr]: 0.09±0.023 UGPx/mg protein) even though it was still a slightly higher than the control group (0.032±0.006 UGPx/mg protein). Furthermore, similar results were observed in their MDA activity level where the oxidative stress had the highest value in the insecticide mixture group and it was significantly decreased with the action of vit. C.”, ”zebrafish juveniles treated with vit. C responded well to the social stimuli from the left arm (pretreatment days: 132.2 ± 11.8 s vs. chronic exposure: 120.5 ± 15.3 s) and had the same pattern as the control group for the center arm (pretreatment days: 56.5 ± 8.05 s vs. chronic exposure: 57.6 ± 7.29 s) and the right arm (pretreatment days: 49.7 ± 6.23 s vs. exposure period: 45.4 ± 9.16 s)” and ”the results demonstrated that at the end of the experiment, vit. C supplement could be implicated in reducing the neurotoxicity of the insecticides in terms of zebrafish social behaviour response, which might indicate the protective function of this vitamin against brain damage.”
- In the discussion: The lack of measurement of vitamin C in the fish or the water needs to be presented as a limitation of the study. Even if this is to be added in future work, the interpretation of data presented is limited by a lack of vitamin C quantification.
As we previously mentioned, the lack of analyzes for vit. C quantification in fish tissues or water is one of our limitation regarding the current study. We were focused on oxidative stress and behavioural abnormalities caused by the insecticides mixture and vit. C supplementation. For sure, there will be further studies which will have this as main target. We added this in the manuscript:”Although we showed here significant results for vit. C supplementation, there are also several limitations of our study as: the lack of measurements regarding quantification of vit. C in water and fish tissues.”.
- Vitamin C is unstable in neutral solutions and can spontaneously oxidize in fish water (resulting in significantly lower levels in a matter of minutes). The result could be the generation of superoxide and/or hydrogen peroxide. Please discuss this in light of your exposure regime. Is there any known research about the interaction with any of these redox compounds with the insecticides used?
We choose the period exposure for vit. C based on literature studies (eg. A. Pyridoxine-Dependent Epilepsy in Zebrafish Caused by Aldh7a1 Deficiency by Pena et al. published in Genetics, 2017; https://doi.org/10.1534/genetics.117.300137.) In addition, we indicated in the discussion section the relation between antioxidants and insecticides mixture and its relevance for our study. The activity of SOD, GPx and MDA had different trends as: ”In agreement with our results, the values (Units/ mg protein) of SOD and GPx at 96 hpf zebrafish embryos were significantly increased after exposure to 0.16-1.66 μg mL-1 pyriproxyfen [17]. Furthermore, another prior study found that the chronic exposure (14 days) of 1.62 µM chlorpyrifos in African tilapia (Oreochromis spilurus) induced lipid peroxidation and alterations in SOD, GSH, and GST levels.”. For our best of knowledge, we do not know any research which had been focused on the interaction of ROS with fipronil and pyriproxyfen.
This manuscript is a resubmission of an earlier submission. The following is a list of the peer review reports and author responses from that submission.
Round 1
Reviewer 1 Report
The authors presented a manuscript about “Vitamin C Attenuates Oxidative Stress and Brain Damage of Fipronil and Pyriproxyfen Insecticide Chronic Exposure on Zebrafish Juvenile (Danio rerio)”.
The topic seems interesting and well within the aims and scopes of the Journal.
Yet the manuscript is very hard to understand since it is very poorly written. In several cases, I must admit that I did not even understand the real meaning of the sentences and, for this reason, I cannot fully judge the content and the scientific soundness of the manuscript itself.
I cannot avoid to recommend the Rejection of this manuscript in order to give the authors the chance to revise it all in the writing part and correct all the grammar and language mistakes.
After this, the authors can resubmit the paper to this same Journal and so let me judge it in the other parts.
Anyway, there are already some things that the authors must check and see:
TITLE:
- Please write the complete name of the species here.
ABSTRACT:
- Please write the complete name of the species here.
KEYWORDS:
- I think a keyword about Danio rerio is necessary here.
INTRODUCTION:
- Line 40: ASD? Please specify this acronym here, too.
- Please write something about the composition of Fip and Pyr.
- Lines 58-59: All the species names must be written in Italics and completely.
- A small description and image of this zebrafish would be interesting to be inserted. Some readers might have never encountered it.
MATERIALS AND METHODS:
- Please name the local pharmacy where you bought Vitamin C. Moreover, are you sure of its purity etc? Please specify the producer, confectioner and other, if possible.
- Please name the store where the insecticide was purchased and how and who performed the quality certification, if possible.
RESULTS:
- How do you explain all the highest and lowest values in those specific days which are quite distant from each other in several cases? Any external cause?
- How do you explain the shortest time spent in moving if compared to pretreatment days in the last two days as reported in lines 216-218.
DISCUSSION:
- The comparisons with the other studies must also report the direct values and be better developed and described.
LANGUAGE, WRITING, GRAMMAR:
- I really suggest the authors to have the manuscript checked by a English native speaker.
Reviewer 2 Report
The research paper attempts to determine the effects of vitamin C on the oxidative harm caused by two insecticides. In order to determine the effect of ascorbic acid, another arm is needed in this study that would include only pure ascorbic acid as there is no way to know that any of the effects seen were due to ascorbic acid alone. The active used is poorly described as a children's liquid vitamin C supplement. No label, supplement name, manufacturer, expiration date or any other detailed information is provided. Thus, this experiment cannot be replicated by any other researcher.
Reviewer 3 Report
This manuscript looks at vitamin C administration as a means to block the harmful effects of insecticides in Danio rerio. Unfortunately, as written the findings in this manuscript are questionable. The main concern is the delivery of vitamin C to the zebrafish and the lack of supporting data to show the administration was effective. This manuscript requires additional experiments and/or supporting data to be acceptable for publication.
Although not an exhaustive list, here are some major concerns:
- No attempt was made to determine how much vitamin C exists in the animals at the study start. No vitamin C content of the diet was provided. This is very important information in this type of study.
- Why was water delivery chosen as a delivery route? This makes it completely irrelevant to terrestrial animals.
- The delivery of vitamin C was questionable. How stable was vitamin C in the water and in the purchased product? Without any validation that vitamin C was indeed added to the water, there is no way of determining that vitamin C was delivered to fish.
- How did vitamin C administration change the chemistry of the fish water, including pH?
- Was "glycerin, propylene glycol and orange flavor" added to control arms of the study? Propylene glycol in particular is a known modulator of neurobehavioral activity in zebrafish models and cannot be ignored.
- How were the insecticides added to the water?
Other Concerns:
- In general, the writing is of inconsistent quality. Sentences are long and grammar is not consistent. The Introduction is very light on vitamin C work in zebrafish - plenty of studies have been done with this model and this vitamin and other antioxidants. Also, the focus on ASD is a little long given the scope of the study. Some of this information should be put in the Discussion; The Discussion is also not well organized.
- The statistics interspersed in the text of the Results makes the text very difficult to read.